# Use of complete medication history to identify and correct transitions-of-care medication errors at psychiatric hospital admission

Victoria Vargas[1], Weston W. Blakeslee[2], Colin A. Banas[2], Christian Teter[1], Katherine Dupuis-Dobson[1], Carol Aboud[1]*

**1** Department of Pharmacy, McLean Hospital, Belmont, Massachusetts, United States of America, **2** Applied Clinical Research Division, DrFirst.COM, Inc., Rockville, Maryland, United States of America

* caboud@partners.org

**Data Availability Statement:** All relevant data are within the paper and its Supporting information files.

## Abstract

Methods for categorizing the scale and severity of medication errors corrected by pharmacy staff during admission medication reconciliation using complete medication history continue to evolve. We established a rating scale that is effective for generating error reports to health system quality leadership. These reports are needed to quantify the value of investment in transitions-of-care pharmacy staff. All medication errors that were reported by pharmacy staff in the admission medication reconciliation process during a period of 6 months were eligible for inclusion. Complete medication history data source was utilized by admitting providers and all pharmacist staff and a novel medication error scoring methodology was developed. This methodology included: medication error category, medication error type, potential medication error severity, and medication non-adherence. We determined that 82 medication errors were detected from 72 patients and assessed that 74 of these errors may have harmed patients if they were not corrected through pharmacist intervention. Most of these errors were dosage discrepancies and omissions. With hospital system budgets continually becoming leaner, it is important to measure the effectiveness and value of staff resources to optimize patient care. Pharmacists performing admission medication reconciliation can detect subtle medication discrepancies that may be overlooked by other clinician types. This methodology can serve as a foundation for error reporting and predicting the severity of adverse drug events.

## Background

Despite advancements in the use of technology at the point of care, medication errors remain prevalent and can occur at every step of the care continuum. Up to 70% of patients have errors on their medication list when admitted to a hospital and up to 59% of errors can cause harm [1]. Medication errors that occur after a patient has been admitted to the hospital may stem from discrepancies in the home medication list [2] and can account for 85% of all inpatient order errors [3]. These medication errors can lead to adverse drug events that may result in

**Funding:** Victoria Vargas, Christian Teter, Katherine Dupuis-Dobson, and Carol Aboud are all current or former salaried employees of McLean. Weston Blakeslee and Colin Banas are salaried employees of DrFirst.com, Inc.

**Competing interests:** Weston Blakeslee and Colin Banas have read the journal's policy and have the following competing interests: They are current or former employees of a health information technology company who are interested in the success of the company.

**Abbreviations:** AHRQ, Agency for Healthcare Research and Quality; CNS, Central Nervous System; EMR, Electronic Medical Record; HIE, Health Information Exchange; LABA, Long Acting Beta Agonist; LOS, Length of Stay; PACT, Program of Assertive Community Treatment; PCNS, Pediatric Clinical Nurse Specialist; PTA, Prior To Admission; TOC, Transitions of Care.

patient harm [4–6]. Medication reconciliation is an established effort to correct these medication errors, especially during transitions-of-care (TOC), but can be time consuming and error-prone [1, 4, 6, 7].

In a psychiatric facility, this process is even more challenging due to acute conditions at admission [8], medication non-adherence [8], incomplete medication history records [9–11], and in some cases, low health literacy [9, 12]. Efforts to correct these medication errors during TOC in a psychiatric setting are continuously improving, but processes and methods for identifying, correcting, and measuring the potential severity of these errors remain poorly defined.

In recent years, there has been a transition to include pharmacy staff in the admission medication reconciliation process and this has been met with pronounced results [9, 13–16]. Admission medication history errors per patient significantly decreased from usual care for patients admitted from the emergency department (8.0±5.6) to medication history collected by pharmacists (1.4±1.9) and pharmacy technicians (1.5±2.1) [5, 17].

Uncovering medication discrepancies is important in all care settings, but especially during TOC. Though the National Coordinating Council for Medication Error Reporting and Prevention (NCCMERP) categorization is routinely used [18–20], there is a lack of standardization when categorizing medication errors that were corrected during TOC. Assessing the value of interventional programs to correct and reduce medication errors during TOC and quantifying the potential harm that was avoided remains a challenge. This study was undertaken to assess the benefit of a previously undescribed method for determining the scale and impact of psychiatric hospital pharmacist error correction through medication reconciliation during the TOC process. We provide a method for addressing this challenge and for estimating the potential impact of pharmacist intervention in correcting medication errors.

## Methods

A retrospective quality improvement observational study was conducted. The study site was an urban, psychiatric care hospital with approximately 324 licensed beds and 218 staffed beds. Discrepancies identified by pharmacists resulting in medication order change were manually recorded. The policy of the Department is to complete electronic documentation for medication discrepancies that pertain to a specific patient and specific error. The DrFirst MedHx web-based platform for medication history data (Rockville, MD) was made accessible to both psychiatric hospital admitting prescribers and pharmacists. After initial use by the admitting prescriber, pharmacists search for outpatient medication history in DrFirst MedHx to identify discrepancies in prior-to-admission (PTA) medications and inpatient medication orders (Fig 1). From August 1, 2019 to February 28, 2020, 82 medication errors from 72 patients (adults aged 18–85) admitted were scored on the type of medication error that was identified and the predicted severity of the error that was avoided.

### Inclusion criteria

All patients were eligible for inclusion and all medication errors that were reported by pharmacists via electronic documentation were collected and organized into a medication error scoring sheet (S2 Appendix—Medication Collection Form). All inpatients with an identified medication discrepancy, as defined by a medication order that deviates from the patient's home medication regimen, were eligible for inclusion. These discrepancies included: dose, frequency, omission, commission, formulation, and substitution. Since this analysis focused on inpatient medication reconciliation, patients that were part of the outpatient Program of Assertive Community Treatment (PACT) were not included.

# DrFirst Medication Reconcilliation Intervention Study Timeline

**Fig 1. Intervention timeline.** (A) AHRQ score and medication discrepancy severity score are to be determine based upon projected course of patient case; choose category and severity based upon the worst case scenario that could occur if the discrepancy was not caught during the hospital length of stay (T1 to T2); medication non-adherence is to be chosen if the use of DrFirst captured medication non-adherence prior to admission; medication error type is to be chosen based upon the initial medication discrepancy identified via the use of DrFirst.

## Medication error data collection

Pharmacists enter information in an iVent, the Epic EMR pharmacy tool used to communicate and record clinical activity, recommendations, and interventions. This is a convenient way to track metrics for medication error documentation, TOC discrepancies, and is only available and visible to pharmacy staff [21–23]. The role of the TOC pharmacist is to review a patient's medication list at admission, regularly in the inpatient setting, and at discharge, correcting any identified discrepancies at the time of documenting errors. More evidence for this type of workflow improving medication reconciliation continues to emerge, but is not ubiquitous in the United States [3, 24].

## Medication error scoring methodology

Our methodology for scoring medication errors is depicted in Table 1. This novel methodology was designed to be an intuitive process to gauge the potential harm of the medication error. We also sought to categorize the type of prevented medication error, assess potential

**Table 1. Medication error classification framework.**

| Measure | Source | Rater Options with Operational Definitions | |
|---|---|---|---|
| Medication Error Category (AHRQ Score) (Projected category of DrFirst case; choose the category based upon the worst case scenario that could take place if the error was not caught) | National Coordinating Council for Medication Error Reporting and Prevention (NCC MERP) | **Category** | **Subcategories** |
| | | No error: No error prevented by the intervention | Category A: Circumstances or events that have the capacity to cause error |
| | | Error, No Harm: An error that could not be expected to cause harm was prevented by the intervention | Category B:An error occurred but the error did not reach the patient ("errors of omission" does reach the patient) Category C: Error occurred, did not cause harm Category D: Error occurred, required monitoring |
| | | Error, Harm: An error that could be expected to cause harm was prevented by the intervention | Category E: Error occurred, caused temporary harm or required intervention Category F: Error occurred, required hospitalization Category G: Error occurred, may have caused permanent harm Category H: Error occurred, intervention required to sustain life |
| | | Error, Death: An error that was expected to cause death was prevented by the intervention | Category I: Error occurred, may have caused death |
| | | N/A | N/A: Unable to identify a potential category |
| | | Harm: Impairment of the physical, emotional, or psychological function or structure of the body and or pain resulting therefrom Monitoring: To observe or record relevant physiological or psychological signs Intervention: May include change in therapy or active medical/surgical treatment Intervention necessary to sustain life: Include cardiovascular and respiratory support (e.g., CPR, defibrillation, intubation, etc.) | |
| Medication Error Type | Adapted from: *Pippins et al. J Gen Intern Med (2008)* *Tam et al. CMAJ (2005)* | Commission: Addition of a drug not used before admission Dose: Error in ordered dose compared to dose used before admission Formulation: Error in ordered formulation compared to formulation used before admission (immediate release versus extended release) Frequency: Error in ordered frequency compared to frequency used before admission Omission: Deletion of a drug used before admission Route: Error in ordered route compared to route used before admission Substitution: Error in ordered medication within the same medication class compared to medication used before admission that is not an intentional substitution due to formulary preference Other: Any error that does not meet any of the above options | |
| Potential Medication Error Severity | Adapted from: *Pippins et al. J Gen Internal Med (2008)* | Significant: An error can cause symptoms yet pose little or no threat to patient's function Serious: An error can cause signs/symptoms associated with serious level of risk that is not life-threatening Life-threatening: An error can cause signs/symptoms that would put patients at risk of death | |

*(Continued)*

**Table 1.** (Continued)

| Measure | Source | Rater Options with Operational Definitions |
|---|---|---|
| Medication Non-adherence | Adapted from:<br>*Pippins et al. J Gen Internal Med (2008)*<br>*Schenis et al. Addict Bebay (2018)* | <u>Completely non-adherent:</u><br>Identified patient is not refilling medication at all<br><u>Sporadically non-adherent:</u><br>The identified patient is refilling medication but at a rate that is slower than necessary for full adherence with no further information from patient<br><u>Systematically non-adherent:</u><br>The identified patient is refilling medication, but at a rate that is slower than necessary for full adherence, with endorsement from the patient that he/she is taking the medication differently than prescribed (e.g., always takes medicine once a day instead of 3 times a day)<br><u>Misuse:</u><br>identified patient identified patient was using medication without a valid prescription/physician's instruction order or using a medication in greater amounts, more often, or longer than per a valid prescription/physician's instruction<br><u>N/A:</u><br>non-adherence not identified |

medication error severity, and categorize medication non-adherence. To minimize subjectivity, the definitions were designed to be simple and inclusive. VV, CAB, and WWB all independently used this methodology to score the medication errors, combined results, and discussed error type until consensus was achieved. Multiple medication errors per patient were separated and scored individually. For gauging harm of the individual medication errors, the authors posed the question: are these medication errors causing harm in the period of admission (i.e. during the patient's stay in the hospital if that stay was the known average length of stay (LOS))? For assessment of long-term effects from a medication error post-discharge, the authors assumed that a psychiatric hospital would correct psychiatric medications prior to discharge that would cause long-term effects had they not been corrected.

The authors recognize that there are multiple uses for certain types of medications that could be considered CNS agents, such as beta blockers being used to treat blood pressure as well as anxiety or akathisia in psychiatric hospitals. The authors carefully considered the indication, dose, and administration of these medications, and all medications that were not indicated for psychiatric treatment were included in post-discharge implications (Table 8).

## Patient demographic data collection

For all 72 patients where a prevented medication error was recorded, patient demographics were collected as outlined in S1 Appendix—Data Collection Tool. Twenty-four distinct data points were collected for every patient in an effort to glean potentially meaningful insights around patient characteristics where medication errors were present and document potential social determinants of health.

## Ethics statement

Our study was reviewed and approved by the Partners Human Research Committee Institutional Review Board (IRB) process under Protocol #: 2020P000670, PI: Aboud, Carol. Written or verbal consent was waived by the IRB due to the difficulty or impossible nature of obtaining informed consent for retrospective medical record review for discharged patients. All identifiable data was stored securely with access limited to study staff, and information resulting from

**Table 2. Patient Demographics.**

| Patient Demographics | Mean / Percentage / SD |
|---|---|
| Female Sex [n (%)] | 41 (57%) |
| Mean Age [years (SD)] | 43.9 (16.4) |
| Mean Length of Stay [Days (SD)] | 13.1 (8.9) |
| Mean Number of Meds at Admission [n (SD)] | 9.2 (6.4) |
| Mean Number of Active Medical Problems | 8.4 (6.4) |

this study will not have important health/medical implications for subjects. Only de-identified data was shared with non-hospital staff co-workers for the purpose of data analysis.

## Results

During the 6-month study period, all inpatients with an identified medication discrepancy in an iVent were eligible for inclusion for analysis by the investigators. Seven patients and 8 potential discrepancies were excluded from the analysis because they were deemed to not qualify as a medication error by the investigators as described in the methods above. Additionally, 2 patients with 3 potential discrepancies were excluded because they were treated in the outpatient PACT program and therefore did not meet inclusion criteria. Consequently, 82 medication errors from 72 patients were included in the analysis. As shown in Table 2: 57% of the patients included in the analysis were female (41/72). The most prevalent discharge diagnoses were: major depressive disorder (21), varying bipolar disorders (13), and schizoaffective disorder (8) (Table 3).

From the 72 patients included in the analysis, the mean age was 43.9 years (SD 16.4 years). The mean LOS was 13.1 days (SD 8.9 days). The mean number of PTA medications was 9.2 (SD 6.4 medications). The mean number of active medical problems was 8.4 (SD 6.4)

**Table 3. Discharge Diagnosis.**

| Diagnosis Type | Sum of Count of Discharge Diagnosis |
|---|---|
| Adjustment Disorder | 1 |
| Affective Psychosis, Bipolar | 2 |
| Alcohol Use Disorder | 5 |
| Anxiety Disorder, Unspecified | 1 |
| Anxiety/Depressive Disorder | 1 |
| Bipolar Disorder, Other | 5 |
| Bipolar I Disorder | 6 |
| Bipolar II Disorder | 2 |
| Dissociative Identity Disorder | 1 |
| Major Depressive Disorder | 21 |
| Panic Disorder | 1 |
| Psychosis | 5 |
| PTSD (Post-Traumatic-Stress-Disorder) | 4 |
| Schizoaffective Disorder | 9 |
| Severe Benzodiazepine Use Disorder | 1 |
| Suicidal Ideation | 2 |
| Unspecified Mental Health Problem | 5 |
| Grand Total | 72 |

**Table 4. Location of Residence.**

| Type of residence | Count of Location of Residence |
|---|---|
| Assisted Living | 3 |
| Group Home | 3 |
| Home | 58 |
| Other | 7 |
| Unknown | 1 |
| Grand Total | 72 |

(Table 2). The vast majority of patients lived at home (58/72), with 3 patients living in a group home and 3 patients in an assisted living setting (Table 4).

From the 82 medication errors documented, 6 different types of medication errors were characterized and scored (Table 5) and described in the Methods Section as well as Table 1. The majority of errors were Dose (32.9%) and Omission (25.6%), followed by Frequency (19.5%), Formulation (12.2%), Commission (4.9%), and Substitution (4.9%).

Though there were a large concentration of medication errors from psychiatric drugs, the drug classes in which errors were found ranged far wider than agents that target the central nervous system (CNS) (Table 6). The majority of errors were found in antidepressants (13/82), followed by anticonvulsants (8/82) and antipsychotics (7/82). The remaining medication errors identified most commonly included the following drug classes: beta2-adrenergic agonist and long-acting corticosteroid combination inhalant (6/82), beta blockers (6/82), and contraceptives (4/82).

From simplifying the Agency for Healthcare Research and Quality's (AHRQ) criteria for harm from a medication error (Table 1), the authors found that 8 errors (8/82) were indeed medication errors, but would not have resulted in harm to the patient during an admission with a typical LOS. Seventy-four of these medication errors may have caused preventable harm to the patient (90.2%) as assessed by the authors based upon the worst-case scenario that could take place if the error was not caught during an admission with a typical LOS (Table 7).

In a further effort to quantify the effect of these medication errors post-discharge, errors were scored on whether they would have post-discharge implications if not corrected (Table 8). Fifty-nine errors that may have been missed had the potential for unintended consequences post-discharge (71.9%).

## Discussion

Quantifying the impact of pharmacist intervention during the medication reconciliation process is a challenge. The goal of this methodology is to create a repeatable, intuitive process to

**Table 5. Types of errors prevented.**

| Medication Error Type | Count of Medication Error Type |
|---|---|
| Commission | 4 |
| Dose | 27 |
| Formulation | 10 |
| Frequency | 16 |
| Omission | 21 |
| Substitution | 4 |
| Grand Total | 82 |

**Table 6. Class of drugs in which error was found.**

| Drug Classes | Count of Medication Class |
|---|---|
| Alpha 1 Agonist | 1 |
| Alpha 1 Blocker | 1 |
| Alpha 2-Adrenergic Agonist | 1 |
| Androgen | 1 |
| Angiotensin-Converting Enzyme (ACE) inhibitor | 1 |
| Antianxiety Agent, Miscellaneous | 2 |
| Antibiotic | 2 |
| Anticonvulsant | 8 |
| Antidepressant | 15 |
| Antidiabetic Agent | 2 |
| Antigout Agent | 1 |
| Antilipidemic Agent | 6 |
| Antimanic Agent | 5 |
| Anti-Parkinsons Agent | 2 |
| Antipsychotic | 5 |
| Beta2-Adrenergic Agonist, Long Acting and Corticosteroid, inhalant (Oral) | 6 |
| Beta-Blocker, Beta-1 Selective | 3 |
| Beta-Blocker, Nonselective | 3 |
| Central Nervous System Stimulant | 1 |
| Contraceptive | 4 |
| Corticosteroid, inhalant (Oral) | 2 |
| Estrogen Derivative | 1 |
| Gastrointestinal Agent, Miscellaneous | 2 |
| Insulin | 1 |
| Mineralocorticoid (Aldosterone) Receptor Antagonist | 1 |
| Nonsteroidal Anti-inflammatory Drug (NSAID), Oral | 1 |
| Ophthalmic Agent, Antiglaucoma | 1 |
| Thyroid Product | 1 |
| Tumor Necrosis Factor (TNF) Blocking Agent | 1 |
| Vitamin | 1 |
| Grand Total | 82 |

**Table 7. Expected harm prevented.**

| Error Severity | Count of AHRQ Score |
|---|---|
| Category B/C/D (Error, No Harm) | 8 |
| Category E/F/G/H (Error, Harm) | 74 |
| Grand Total | 82 |

**Table 8. Post discharge implications were there post-discharge implications.**

| | Count of Post-discharge Implications |
|---|---|
| No | 23 |
| Yes | 59 |
| Grand Total | 82 |

quantify the scale and type of medication errors, as well as give high-level metrics for the severity that they could have caused. The authors determined that classifying errors by potential harm in a worst-case scenario was the most clinically appropriate course of action.

This method proves the importance of utilizing pharmacists in the admission medication reconciliation process and adds clarity to the scale and severity of medication errors that has been previously difficult and cumbersome to measure and convey [25, 26]. A potential next step is to apply this method into a real-time or near real-time dashboard that would easily and quickly convey the findings of pharmacist-mediated medication reconciliation to hospital quality staff and leadership. This method can be used to quantify the extent of prevented medication errors, convey the importance of eliminating TOC medication errors, and institute a process of continual improvement for inpatient medication management. All of these insights are critical for clinical leadership to know when making resource allocation decisions.

Additionally, many of the medication errors in a psychiatric setting were not necessarily psychiatric medications, highlighting the importance of robust medication reconciliation at all settings of care, including: primary care, specialty, inpatient, and emergency visits. The investigators assumed that all medication errors that were categorized under psychiatric medications would be corrected over the course of the patient's stay in the hospital. Despite this assumption, as shown in Table 8, around 72% of errors (59/82) could have had long-term effects post-discharge if not corrected (i.e. statins/metformin). Without this assumption, the extent of post-discharge implications becomes even higher.

When analyzing the types of errors detected by pharmacists, formulation errors particularly stand out. Pharmacists are more in tune to the miniscule differences between medication formulations and dosages. Since 10/82 of these errors were formulation errors and 27/82 were dose related errors, this indicates that having pharmacists review the medication history was impactful. With 74/82 (88%) of the reported medication errors having the potential to harm the patient even after initial use of outpatient medication history by an admitting prescriber, our analyses underscore the importance of pharmacists intervening in the medication reconciliation process.

Omissions were another type of error that stood out. Though there have been many efforts to improve data interoperability in the United States, particularly among home medication lists [27], closing the gap of finding missing medications currently being taken by the patient remains cumbersome and challenging. This is due to long hours spent calling pharmacies, incomplete pharmacy claims data, and laborious interviews with patients and caregivers [28]. 26% of these omissions were identified with more complete medication history data, underscoring the importance of this resource for TOC pharmacists.

Having up-to-date, complete, accurate medication history data is a powerful tool to help reconcile a patient's medication regimen. Oftentimes, patients have inaccurate accounts of their medication history, current medication list, or refill history which results in providers calling pharmacies, calling caregivers, or relying on patients to bring a bag of their medications with them in order to get a better sense of the patient's current medication regimen [29, 30]. Medication history tools that include records of prescriptions from pharmacy benefit manager claims, local and independent pharmacy transactions, health information exchanges (HIEs), EMR data, and electronic prescription records serve as more complete, transactional resources. These give clinicians a more comprehensive and unbiased medication history to help guide these important conversations with patients. Additionally, these tools eliminate the need for physical review of medication bottles in an inpatient setting and the associated storage and diversion liability. Based on our findings, the combination of the medication expertise of a pharmacist and the use of a medication history tool can decrease TOC medication errors thereby improving patient care.

We have identified several limitations in our study. Four of the authors that are pharmacy staff are passionate about patient care and likely hold confirmation bias that pharmacist intervention in the medication reconciliation process is the most effective way to conduct these methods. Two of the authors are current or former employees of a health information technology company who are interested in the success of the company. Additionally, pharmacists were not directly involved in conversations with patients related to medication use upon admission and therefore did not have the opportunity to assess real-life medication use verses reported data. Also, the potential severity of corrected medication errors was considered based on the worst-case scenario for harm and did not consider the breadth of all possible severity for corrected errors. Conscious efforts were made to remain objective by all authors with these limitations in mind.

Potential follow-up studies may include documenting medication non-adherence and potential barriers to first fill, such as how cost to the patient may impact omission medication errors. These were not directly addressed by TOC pharmacy staff during the time setting of this study, and would be an interesting follow up analysis in psychiatric patients. Prescription abandonment rates dramatically increase as out of pocket cost increases [31–33]. Applying this method to outpatient medication reconciliation programs would also be of interest to quantify the impact of pharmacy staff in the medication reconciliation process and compare this impact to inpatient initiatives. Lastly, a conversation with patient and/or caregivers about real-life medication use is a vital portion of a true medication reconciliation. However, while this is expected on admission, the quality of this type of conversation was not part of our intervention and would be an important aspect of a future study.

## Conclusions

We believe that this method is of particular importance for complex patients because increased complexity in the medication regimen generally leads to lower adherence, less accurate recollection of medications, and poorer overall health. Our results support the value of pharmacist expertise while utilizing a medication history tool for TOC medication reconciliation in a psychiatric hospital setting, as well as other non-psychiatric settings [34, 35]. Additionally, we have provided a detailed, yet user-friendly methodology and several convenient tools for clinicians to utilize when scoring the types of medication errors and predicting the severity of these errors that were intercepted by pharmacist intervention.

## Supporting information

**S1 Appendix. Data collection tool.** Data table describing the source of specific data elements and their formats to be included in the medication error collection form.
(DOCX)

**S2 Appendix. Medication error collection form.** Blank medication error data collection sheet to be used by pharmacy TOC staff.
(XLSX)

**S1 Dataset. Minimum underlying dataset.** De-identified medication error scoring results used by the authors.
(XLSX)

## Acknowledgments

The authors would like to acknowledge McLean pharmacists for their work in identifying and reporting these transitions of care medication discrepancies.

We would also like to thank Yan Zhuang, RN, BSN for her administrative assistance and background knowledge on this study.

## Author Contributions

**Conceptualization:** Victoria Vargas, Colin A. Banas, Christian Teter, Katherine Dupuis-Dobson, Carol Aboud.

**Data curation:** Weston W. Blakeslee, Christian Teter, Katherine Dupuis-Dobson, Carol Aboud.

**Formal analysis:** Victoria Vargas, Weston W. Blakeslee, Colin A. Banas.

**Investigation:** Victoria Vargas, Weston W. Blakeslee, Colin A. Banas, Katherine Dupuis-Dobson, Carol Aboud.

**Methodology:** Victoria Vargas, Weston W. Blakeslee, Colin A. Banas, Christian Teter, Katherine Dupuis-Dobson, Carol Aboud.

**Project administration:** Victoria Vargas, Weston W. Blakeslee, Christian Teter, Carol Aboud.

**Writing – original draft:** Victoria Vargas, Weston W. Blakeslee, Colin A. Banas, Carol Aboud.

**Writing – review & editing:** Victoria Vargas, Weston W. Blakeslee, Colin A. Banas, Christian Teter, Katherine Dupuis-Dobson, Carol Aboud.

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
