## [Decision Letter · Decision Letter 0]

5 Jul 2022

PONE-D-22-13219Use of Complete Medication History to Identify and Correct Transitions-of-Care Medication Errors at Psychiatric Hospital AdmissionPLOS ONE

Dear Dr. Aboud,

Thank you for submitting your manuscript to PLOS ONE. After careful consideration, we feel that it has merit but does not fully meet PLOS ONE’s publication criteria as it currently stands. Therefore, we invite you to submit a revised version of the manuscript that addresses the points raised during the review process.

We look forward to receiving your revised manuscript.

Kind regards,

Vijayaprakash Suppiah, PhD

Academic Editor

PLOS ONE

Journal Requirements:

“No funding was exchanged by McLean or DrFirst, Inc. Both organizations provided the labor for this study at their own cost.”

5. Please amend your manuscript to include your abstract after the title page.

6. Please upload a copy of Figure 2 and 3b, to which you refer in your text on page 7, 8 and 9. If the figure is no longer to be included as part of the submission please remove all reference to it within the text.

Reviewers' comments:

Reviewer's Responses to Questions

**Comments to the Author**

1. Is the manuscript technically sound, and do the data support the conclusions?

Reviewer #1: Partly

Reviewer #2: Yes

2. Has the statistical analysis been performed appropriately and rigorously? 

Reviewer #1: Yes

Reviewer #2: Yes

3. Have the authors made all data underlying the findings in their manuscript fully available?

Reviewer #1: Yes

Reviewer #2: Yes

4. Is the manuscript presented in an intelligible fashion and written in standard English?

Reviewer #1: Yes

Reviewer #2: Yes

5. Review Comments to the Author

Reviewer #1: Thank you for submitting this well written and carefully considered study addressing a very pertinent research question.

Introduction

The introduction was well written and supported the need for the research.

Methods

The methods did not stipulate that the study was conducted at a psychiatric hospital/ward despite the introduction indicating so. I would like to see a sentence providing more background on the nature of the hospital – e.g number of beds, location (i.e rural vs metropolitan), staffing etc would provide context to the significance of these 82 identified errors in 6 months.

The authors mentioned “identified, legitimate medication discrepancy (Lines 125-127)” – what would constitute to it being legitimate or illegitimate? Please elaborate.

It would also be of value to the readers for the authors to briefly summarise the role of the pharmacist in this context. In particular, how do they usually identify these discrepancies in their daily practice? Is there an established process or is this what the research is trying to initiate?

Results

Table 1: I noted that in the row “Medication Error Type,” Other is listed in the middle of the list. It may be better for other to be listed at the bottom of the list given that the description is “any error that does not meet any of the above options.”

Table 2d: I noted that one of the (2nd) most common medication error type is “Omission.” Does this suggest that a best possible medication history was not collected (in Australia, this is the role of the pharmacist). I would like to see this explored in the discussion.

Discussion

The discussion does not contain any references to the existing literature. I would like to see the findings be discussed in the context of the current body of literature. For example, “…adds quantification and clarity to the scale and severity of medication errors that has been previously difficult and cumbersome to measure and convey (Lines 212-215).” This would need a reference.

I feel that the results can be explored/explained more in the discussion section. There is a brief discussion about formulation errors (line 231), however, I feel that other errors such as frequency and omission can also be further elaborated.

Multiple uses for certain types of medications (Lines 240-245): would this be part of the methods?

Lines 273-275: “potential follow up studies may include documenting medication non-adherence and potential barriers to first fill such as cost to the patient.” I cannot see the benefit of this or how this would add to the findings presented. Please further elaborate.

Conclusion

Repetition: Lines 285-287 and 287-289.

The authors claim to “prove the value of pharmacist expertise while utilizing a medication history tool for TOC medication reconciliation.” Please elaborate how the authors were able to make this conclusion based on the results presented (i.e 82 medication errors in six months, how does this compare to the existing literature). Would other health professionals such as nurses be able to identify such errors? Maybe the skills and expertise of the pharmacists have been shown/proven in other studies. This could be discussed and referenced in the discussion if appropriate.

Overall

I find the manuscript well written and interesting, addressing a pivotal research question. I am concerned that the discussion is not supported by references. I would suggest some restructuring to the discussion, especially to include references to other relevant/similar research. For example, “oftentimes, patients have inaccurate accounts of their medication history, current medication list or refill history…” is this based on personal experience?

Reviewer #2: It's an interesting study so in my opinion can publish it as it .

It has a good point for the pharmacists role in our hospitals and has a good idea to protect patients from drug drug interaction or overlaps or ovser doses

.

6. PLOS authors have the option to publish the peer review history of their article (what does this mean?). If published, this will include your full peer review and any attached files.

Reviewer #1: No

Reviewer #2: No

---

## [Author Response · Author response to Decision Letter 0]

17 Aug 2022

To our reviewers and editors at PLOS One, thank you for your constructive comments. Please find our responses to each comment below:

• We have included an Ethics Statement in the Methods section of our manuscript at the direction of the editor.

• We have added additional clarification to our financial disclosure to our cover letter at the direction of the editor.

• We have included our updated Data Availability Statement in our cover letter. Also, we have uploaded our Minimum Underlying Dataset as Supporting Information and will include the following statement in our manuscript: All data is included in the manuscript and/or Supporting Information. We added legends for all Supporting Information and named the files to be consistent with PLOS’ submission guidelines.

• Instead of breaking out the Abstract into separate Objective, Methods, Results, and Conclusions, we have amended the manuscript to include the Abstract in one cohesive paragraph.

• We double checked to ensure that none of the papers we cited have been retracted. All our citations correctly refer to the preceding statements.

Reviewer Comments Addressed:

• We have provided a sentence at the beginning of the methods section to provide more context on the nature of the hospital.

• We have clarified what a “legitimate medication discrepancy” is in the methods section by removing the word “legitimate” and describing a medication discrepancy.

• We have added a brief summary of the role of a TOC pharmacist in the context of transitions of care and supporting citations for the benefits of this clinical workflow.

• We have moved the Medication Error Type of “Other” to the bottom of the list instead of the middle where it makes more sense to display.

• We explored the 2nd most common medication error we found (Omission) in the discussion as requested.

• We agree that the methodology for scoring medication that have multiple uses belongs in the methods section and has been moved there.

• We have added references that support our claims of the difficulty to quantify the scale and severity of medication errors. 

• We deleted the second repetitive sentence identified in the conclusion.

• In support of our claim that we “proved the value of pharmacist expertise while utilizing a medication history tool for TOC medication reconciliation,” we have added references and clarification adjacent this claim, changed the word ‘proved’ to ‘support’, and we also introduced this concept with this sentence earlier in the discussion “Pharmacists are more in tune to the miniscule differences between medication formulations and dosages”

• We added the following clarifying sentence to address how a follow up study measuring medication cost may add to the omission findings presented “These were not directly addressed by TOC pharmacy staff during the time setting of this study and would be an interesting follow up analysis to measure the effect of prescription drug costs on omission medication errors in psychiatric patients.”

• We have included citations to support the claim “oftentimes, patient have inaccurate accounts of their medication history, current medication list, or refill history” in the discussion.

Font size

• Standard font was used in the manuscript

• Major heading were formatted In bold 18pt font

• Sub heading were formatted in bold 16pt font

Table formats

• Table title and caption was inserted after paragraph of initial mention

• Actual table itself was inserted after paragraph of initial mention

• Tables were converted into editable cells

• 1pt font was used for border

• 3pt font was used for lever

• Only black color font was used

Any other changes, etc.

• Figure title and caption was inserted after paragraph of initial mention

• Figure is provided separately

• Acknowledgements were added before references

• Supporting files are added after references

• References are recited into correct format

• Retractable references checked (none noted)

We hope that these revisions satisfy the reviews and eagerly await your decision.

---

## [Decision Letter · Decision Letter 1]

6 Oct 2022

PONE-D-22-13219R1Use of complete medication history to identify and correct transitions-of-care medication errors at psychiatric hospital admissionPLOS ONE

Dear Dr. Aboud,

Thank you for submitting your manuscript to PLOS ONE. After careful consideration, we feel that it has merit but does not fully meet PLOS ONE’s publication criteria as it currently stands. Therefore, we invite you to submit a revised version of the manuscript that addresses the points raised during the review process.

We look forward to receiving your revised manuscript.

Kind regards,

Vijayaprakash Suppiah, PhD

Academic Editor

PLOS ONE

Journal Requirements:

Reviewers' comments:

Reviewer's Responses to Questions

**Comments to the Author**

1. If the authors have adequately addressed your comments raised in a previous round of review and you feel that this manuscript is now acceptable for publication, you may indicate that here to bypass the “Comments to the Author” section, enter your conflict of interest statement in the “Confidential to Editor” section, and submit your "Accept" recommendation.

Reviewer #1: All comments have been addressed

2. Is the manuscript technically sound, and do the data support the conclusions?

Reviewer #1: Yes

3. Has the statistical analysis been performed appropriately and rigorously? 

Reviewer #1: N/A

4. Have the authors made all data underlying the findings in their manuscript fully available?

Reviewer #1: Yes

5. Is the manuscript presented in an intelligible fashion and written in standard English?

Reviewer #1: Yes

6. Review Comments to the Author

Reviewer #1: Thank you for addressing the comments. I have some minor comments.

Abstract

The abstract could benefit from further editing, I noticed that there were frequent uses of long sentences which can reduce the readability of the paper.

Introduction

I do not feel that ‘feasibility’ is the appropriate terminology for what was achieved in this study. The study seems to do the following:

1. Design a tool to assess and quantify the risk of medication errors

2. Assess the benefit of TOC pharmacists in identifying medication errors

If this is the case, authors should reconsider the wording to better reflected what was achieved in this study. The abstract mentions “Establishing a method for categorizing the scale and severity of medication errors…” which I think is a better reflection of the study’s aim.

Methods:

Inclusion criteria:

This section would benefit from some minor editing. For example Lines 155-159 is long and cumbersome.

Results

Format – line 216-217, needs a space between the table and the next paragraph.

Discussion

I feel that the discussion will benefit from further editing. In particular, sentences could be shorten to improve readability.

Figure: DrFirst Medication Reconciliation Intervention Study Timeline

The figure I received was of poor quality and was difficult to read as a result. A better quality figure/image may be needed for publication. Unless this is managed by the journal?

Overall

The manuscript highlights the importance of the TOC pharmacists and addresses a key area. I would suggest some minor revision to the manuscript, especially the discussion, to improve readability. In particular, sentences should be shortened and made more concise.

7. PLOS authors have the option to publish the peer review history of their article (what does this mean?). If published, this will include your full peer review and any attached files.

Reviewer #1: No

---

## [Author Response · Author response to Decision Letter 1]

16 Nov 2022

To our reviewers and editors at PLOS One, thank you for your constructive comments. We are happy to accommodate your additional revisions to improve the readability of our manuscript. Please find our responses to each comment below:

Reviewer Comments Addressed:

Abstract

• We have edited the abstract to eliminate long sentences to improve the readability of the paper.

Introduction

• We have removed the term feasible and feasibility from the manuscript and re-worded our explanations to reflect “assessing the benefit” of our methodology in response to our reviewer’s comments.

Methods:

• We edited the following sections to improve reading clarity: Inclusion Criteria, Medication Error Data Collection, and Medication Error Scoring Methodology.

Results

• We added a space between the table and the next paragraph for lines 216-217.

Discussion

• We significantly edited this section by shortening sentences and breaking out an additional section to improve readability.

Figure: 

• We did a full re-design of the figure ‘DrFirst Medication Reconciliation Intervention Study Timeline’ per the reviewer’s suggestions.

Thank you for the clarifying reviews. We believe they will greatly enhance the readability of our manuscript and eagerly await your response.

---

## [Decision Letter · Decision Letter 2]

19 Dec 2022

Use of complete medication history to identify and correct transitions-of-care medication errors at psychiatric hospital admission

PONE-D-22-13219R2

Dear Dr. Aboud,

We’re pleased to inform you that your manuscript has been judged scientifically suitable for publication and will be formally accepted for publication once it meets all outstanding technical requirements.

Kind regards,

Vijayaprakash Suppiah, PhD

Academic Editor

PLOS ONE

Additional Editor Comments (optional):

Dear authors, 

Please note that the reviewer has highlighted that the figure was very blurry. Please submit the figure with better resolution. 

Thank you. 

Reviewers' comments:

Reviewer's Responses to Questions

**Comments to the Author**

1. If the authors have adequately addressed your comments raised in a previous round of review and you feel that this manuscript is now acceptable for publication, you may indicate that here to bypass the “Comments to the Author” section, enter your conflict of interest statement in the “Confidential to Editor” section, and submit your "Accept" recommendation.

Reviewer #1: All comments have been addressed

2. Is the manuscript technically sound, and do the data support the conclusions?

Reviewer #1: Yes

3. Has the statistical analysis been performed appropriately and rigorously? 

Reviewer #1: N/A

4. Have the authors made all data underlying the findings in their manuscript fully available?

Reviewer #1: Yes

5. Is the manuscript presented in an intelligible fashion and written in standard English?

Reviewer #1: Yes

6. Review Comments to the Author

Reviewer #1: Thank you for addressing the comments. I noted that figure 1 has been greatly improved since the first submission. However, my copy was very blurry, I think the figure may need to be resubmitted (with better resolution) for publication.

7. PLOS authors have the option to publish the peer review history of their article (what does this mean?). If published, this will include your full peer review and any attached files.

Reviewer #1: No

---

## [Editor Report · Acceptance letter]

2 Jan 2023

PONE-D-22-13219R2 

Use of complete medication history to identify and correct transitions-of-care medication errors at psychiatric hospital admission 

Dear Dr. Aboud:

I'm pleased to inform you that your manuscript has been deemed suitable for publication in PLOS ONE. Congratulations! Your manuscript is now with our production department. 

Kind regards, 

on behalf of

Dr. Vijayaprakash Suppiah 

Academic Editor

PLOS ONE